# Genome-Wide Identification and Expression Analysis of *CsCaM/CML* Gene Family in Response to Low-Temperature and Salt Stresses in *Chrysanthemum seticuspe*

**DOI:** 10.3390/plants11131760

**Published:** 2022-07-01

**Authors:** Manman Fu, Chao Wu, Xia Li, Xiaoyu Ding, Fangqi Guo

**Affiliations:** Institute of Horticulture, Zhejiang Academy of Agricultural Sciences, Hangzhou 310058, China; sunshineabigail@163.com (M.F.); wuchao@zaas.ac.cn (C.W.); lix@zaas.ac.cn (X.L.); dingxy321@163.com (X.D.)

**Keywords:** *Chrysanthemum seticuspe*, calmodulin (CaM), calmodulin-like proteins (CML), gene family, low-temperature, salt stress

## Abstract

Calmodulin (CaM) and calmodulin-like proteins (CML) act as significant Ca^2+^ sensors binding Ca^2+^ with EF-hand motifs and have been reported to be involved in various environmental stresses in plants. In this study, calmodulin *CsCaM/CML* gene family members were identified based on the genome of *Chrysanthemum seticuspe* published recently; a phylogenetic tree was constructed; gene structures and chromosomal locations of *CsCaM/CML* were depicted; cis-acting regulatory elements were predicted; collinearity and duplicate events of *CaM/CML* were analyzed using MCScanX software; and the expression levels of *CsCaM/CML* in response to abiotic stress were analyzed, based on the published RNA-seq data. We identified 86 *CsCaM/CML* (4 *CsCaMs* and 82 *CsCMLs*) genes in total. Promoter sequences of *CsCaM/CML* contained elements related to abiotic stresses (including low-temperature and anaerobic stresses) and plant hormones (including abscisic acid (ABA), MeJA, and salicylic acid). *CsCaM/CML* genes were distributed on nine chromosomes unevenly. Collinearity analysis indicated that recent segmental duplications significantly enlarged the scale of the *CML* family in *C. seticuspe*. Four *CsCMLs* (*CsCML14*, *CsCML50*, *CsCML65*, and *CsCML79*) were statistically differentially regulated under low-temperature and salt stress compared with those in the normal condition. These results indicate diverse roles of *CsCaM/CML* in plant development and in response to environmental stimuli in *C. seticuspe*.

## 1. Introduction

Calcium (Ca^2+^), one of the most significant second messengers of signal transduction, is reported to have been involved in abiotic stress responses as well as plant growth and development [1,2,3]. Ca^2+^ was stimulated by various abiotic stresses, including salt, low-temperature, and oxidative stress [1]. The ability of Ca^2+^ as a messenger, interacting with other proteins that modulate plant stress and plant development, is guaranteed by Ca^2+^ sensors, such as calmodulin (CaM), Ca^2+^ protein kinases (CPKs) or Ca^2+^-dependent protein kinases (CDPK), calcineurin B-like proteins (CBL), and calmodulin-like proteins (CML) [4,5,6].

CaMs are conserved Ca^2+^-binding proteins, typically with four EF-hand motifs and CML, usually with 1–7 EF-hand motifs. Each EF-hand motif consists of two alpha helices which are connected by a 12-amino-acid residue loop [7]. The EF-hand motif can accommodate calcium and magnesium that which have subtle differences in affinity, as well as a wide range of induced conformational changes [8]. *CaM/CMLs* were involved in multiple processes related to plant development, growth, and various environmental stress responses [9]. Magnan et al. found that *AtCML9* was significantly stimulated under ABA and abiotic stress treatments and showed a significant role in salt stress responses in *Arabidopsis thaliana* [10]. *AtCML20* was negatively regulated under drought stress, while overexpression of rice *CML OsMSR2* could enhance the drought tolerance of *A. thaliana* [11,12]. Rao and his team found that overexpressing *GmCaM4* in soybean could enhance resistance to pathogens and salt stress [13]. Townley et al. found that *AtCaM* negatively regulated the expression of *COR* (*Cold on regulated*), *D29A*, *KIN1*, and *KIN2* [14].

With the development of sequencing technology, the genome-wide identification of *CaM/CMLs* was analyzed in monocots (including rice (*Oryza Sativa* L.)), multiple eudicots (including brassicaceae plants such as *A. thaliana*, Chinese cabbage (*Brassica rapa* L. ssp. *Pekinensis*), *Carica papaya*, and *Brassica oleracea*), solanaceae plants (including tomato (*Solanum lycopersicum* and *Solanum pennellii*)), rosaceae plants (including woodland strawberry (*Fragaria vesca*) and apple (*Malus* × *domestica*)), fabaceae plants (including *Lotus japonicas,* as well as vitales plants (including grapevine (*Vitis vinifera*)) [15,16,17,18,19,20,21,22,23,24,25,26,27,28]. Chrysanthemum, one of the most industrially essential cut flowers worldwide, has charmed people with various flower colors and morphologies [29], and is susceptible to long-term cold stress [30]. Previous transcriptomic analysis revealed that five *CML*-resembling genes in *Chrysanthemum nankingense* were significantly changed under cold acclimation [31]. Recently, Nakano et al. constructed a pure diploid, a model strain, and Gojo-0 (*C. seticuspe*) with self-compatibility. They sequenced its genome and obtained a chromosome-level reference genome with 3.05 Gb, covering 97% of the *C. seticuspe* genome [29]. Therefore, it is necessary to perform genome-wide identification and expression analysis of *CaM/CML* genes in *C. Seticuspe* for candidate gene selection and further functional characterization.

In this study, a whole-genome-scale identification of *CaM/CML* genes was performed using CaM/CML members of *A. thaliana* with the accessibility of the chromosomal-level assembling genome of *C. seticuspe* [29]. A phylogenetic tree of CsCaM/CML and AtCaM/CML was constructed. Gene structures, motifs or domains, and cis-acting regulatory elements of promoters of *CsCaM/CML* genes were analyzed. Collinearity analysis of *CaM/CML* genes between *C. seticuspe* and *A. thaliana* was performed. The expression levels of *CsCaM/CML* members in response to cold stress and salt stress were analyzed.

## 2. Results

### 2.1. Identification and Phylogenetic Analysis of CaM/CML in C. seticuspe

A total of 86 *CaM/CML* were identified based on BLASTP and SMART analysis, containing 82 *CML* and 4 *CaM* gene family members. The phylogenetic analysis showed that 86 CsCaM/CML were divided into nine groups, from Group I to Group IX (Figure 1). Group I contained 4 CsCaM members, and 82 CsCMLs belonged to Groups II–IX.

### 2.2. Analysis of Gene Structure, Protein Motif, and Cis-Acting Regulatory Elements of Promoters

The gene structure of *CsCaM/CML* members was analyzed according to their coding and genomic sequences. Most *CsCaM/CML* gene family members in the same group shared similar gene structures (Figure 2a). The number of introns varied from 0 to 10. Almost all the members in Group V-VIII contained no intron except *CsCML79* in Group V, *CsCML11* in Group VI, *CsCML37* in Group VII, and *CsCML32* and *CsCML82* in Group VIII. *CsCaM/CML* members in Group IX had the highest number of introns, generally greater than six. The motifs of CsCaM/CML were discovered by submitting the protein sequences using the online tools. The discovered motifs in the same group almost shared the same motifs (Figure 2b). The prediction of cis-acting regulatory elements of promoters showed that *CsCAM/CML* contained cis-acting regulatory elements associated with abiotic stresses such as low-temperature, drought, and anaerobic stress (Figure 2c); in addition, six elements involved in light response were observed. Elements in response to plant hormone and signaling molecules such as auxin, abscisic acid (ABA), MeJA, and salicylic acid were found in *CsCMLs* promoters (Figure 2c). *CsCML63* contained 13 ABRE elements and 14 elements related to MeJA responses (Figure 2c).

### 2.3. Chromosomal Location and Cis-Acting Regulatory Elements of Promoters

According to the position of *CsCaM/CML* members on the *C. seticuspe* chromosome, chromosomal locations were depicted. Eighty-six *CsCaM/CML* gene family members distributed on nine chromosomes unevenly (Figure 3, Appendix A). Chromosome 8 contained the highest number of *CsCaM/CMLs*, while chromosome 2 contained the lowest number of *CsCaM/CMLs*. *CsCaM1*, *CsCaM2*, *CsCaM3*, and *CsCaM4* were located on chromosome 1, chromosome 2, chromosome 3, and chromosome 4, respectively.

### 2.4. Collinearity of CMLs in C. seticuspe

We obtained 926 collinearity blocks containing 14,449 genes in *C. seticuspe* and *A. thaliana* using collinearity analysis, which included 24 *CMLs* of *C. seticuspe* and 15 *CMLs* of *A. thaliana* (Figure 4a). In total, 469 blocks containing 6970 genes were observed among chromosomes of *C. seticuspe* (Figure 4b). Duplicate events analysis showed that all 4 *CaMs* and 45 *CMLs* in *C. seticuspe* were considered as dispersed duplications; 14 *CMLs* were proximal events; 5 *CMLs* were tandem duplications (Appendix A); and 18 *CMLs* were recognized as whole genome (or segmental) duplications (Figure 4). Enrichment analysis indicated that whole genome duplications (WGD) of *CMLs* were significant with a *p* value of 0.0065.

### 2.5. Expression Analysis of CsCaM/CML in Response to Abiotic Stress

To explore the response of *CsCaM/CML* under abiotic stress, the RNA-seq data were downloaded from the ENA database. The expression levels of *CsCaM/CML* showed variety under cold stress compared with normal conditions (Figure 5A). Fifteen *CsCMLs* were statistically significantly regulated under cold treatments including four significantly up-regulated *CMLs* (*CsCML2*/*3*/*9*/*80*) and eleven significantly down-regulated *CMLs* (*CsCML5*/*14*/*15*/*16*/*18*/*23*/*40*/*50*/*65*/*75*/*79*) (Figure 5B). We found that most of *CaM/CML* genes in *C. seticuspe* were up-regulated under salt stress compared with normal conditions (Figure 6A). Expression-level analysis revealed that 13 *CsCMLs* were statistically significantly regulated in response to salt stress, containing 12 significantly up-regulated *CMLs* (*CsCML14/34/38/45/50/60/65/67/69/70/79/81*) and one significantly down-regulated *CML* (*CsCML56*) (Figure 6B).

## 3. Discussion

A total of 86 *CsCML/CaM* genes, including 82 *CML* members and 4 *CaM* members, were identified using BLASTP with 57 CML/CaM from *A. thaliana* as the query. The phylogenetic analysis showed that those CML/CaM can be divided into nine groups, including one CaM group containing seven AtCaMs and four CsCaMs (Figure 1), which is similar with the finding in previous studies [16,32]. Furthermore, 82 CML members in *C. seticuspe* were divided into eight groups unevenly (Figure 1). Protein motifs of CMLs showed diversity among different groups (Figure 2b). Cis-acting regulatory elements represented variety in *CML* genes promoters in *C. seticuspe*, which indicates the diverse functions of *CMLs* in plants [27]. EF-hand motifs were considered the only motif in CML and played crucial roles in interacting with binding calcium ions [9]. All *CsCML* genes identified in this study contained two or more EF-hands (Figure 2b), indicating they have necessities in binding calcium ions. The identity of amino acid sequences of CsCML and CaM in *Arabidopsis* ranged from 16% to 59% (except CsCML60 with identity from 78% to 92%); the identity of amino acid sequences of CsCML and predicted CaM in *Chrysanthemum nankingense* ranged from 19% to 58% (except CsCML60 with identity larger than 90%). Though CsCML60 was considered as one CML in this study based on the phylogenetic analysis, it might be a CaM considering that it has high identity with CaMs and it contains four typical EF-hand motifs. Moreover, though CsCaM3 has higher identity (> 96%) than CaM in *Arabidopsis*, more work needs to be carried out to explore whether it is a true CaM since it contains two EF-hand motifs and the amino acid sequence is much shorter than others. Despite CsCML60, the identity result was higher than that in *Arabidopsis* (16%) and relatively lower than that found in *Solanum lycopersicum* (24%~79%) [9,21].

A previous study identified recent segmental duplication in *C. seticuspe* [29]. In the present study, we observed 18 *CsCMLs* with whole genome (or segmental) duplications based on collinearity analysis (Figure 4b). Enrichment analysis indicated that WGD or segmental duplications played a significant role (with *p* value of 0.0065) in the expansion of *CML* genes in *C. seticuspe*, which supports the segmental duplication events in *C. seticuspe*. Additionally, collinearity analysis revealed 926 collinearity blocks between *C. seticuspe* and *A. thaliana* involving 24 *CMLs* in *C. seticuspe* and 15 *CMLs* in *A. thaliana* (Figure 4a), which is consistent with previous studies [27,32]. These results indicate the conservativeness and potential role of *CML* genes in plants adapting to the environment.

*CMLs* were reported to be involved in the response to environmental stresses. Knocking out *AtCML9* could enhance the tolerance to drought and salinity in *A. thaliana* [10]. Vanderbeld and his co-workers analyzed the expression of *CML* in *A. thaliana* with promoters, and observed the different stimulations of *AtCML37* and *AtCML38* under salt, oxidative, and drought stress, as well as hormonal treatments [33]. In this study, we analyzed cis-acting regulatory elements of upstream sequences (2000 bp) of *CaM/CML* in *C. seticuspe* and observed various elements involved in abiotic stress (such as low-temperature, drought, and anaerobic stress) and plant hormone response elements such as auxin, ABA, MeJA, and salicylic acid (Figure 2c). Several *CMLs* contained multiple cis-acting regulatory elements in response to stress, for example, *CsCML63* which contained 13 ABA-related elements (ABRE) and 7 elements associated with MeJA responsiveness (TGACG-motif) simultaneously (Figure 2c). This indicated their diverse functions in response to multiple environmental stress conditions in *C. seticuspe*. Expression analysis showed that 15 *CMLs* were statistically differently regulated under cold stress, containing 11 down-regulated *CMLs* (*CsCML5*/*14*/*15*/*16*/*18*/*23*/*40*/*50*/*65*/*75*/*79*) and 4 up-regulated *CMLs* (*CsCML2*/*3*/*9*/*80*); 13 *CML* genes (including 12 up-regulated genes (*CsCML14*/*34*/*38*/*45*/*50*/*60*/*65*/*67*/*69*/*70*/*79*/*81*) and a down-regulated gene (*CsCML56*)) showed statistically different expressions in *C. seticuspe* under salt stress compared with normal conditions (Figure 5 and Figure 6). Previous publications provided evidence of *CML* genes in response to diverse stimuli. Delk’s team found that *AtCML24* showed responses to ABA, salt treatments, different daylengths, and long-day-induced transition to flowering [34]. The expression of *SlCML26* in tomato (*Solanum lycopersicum*) was significantly regulated under cold, drought, and salt stress [21]. Expressions of four *CsCMLs* (*CsCML14*, *CsCML50*, *CsCML65*, and *CsCML79*) were statistically regulated under cold and salt stress (Figure 5 and Figure 6), which indicates diverse functions of these *CsCMLs* under diverse environmental stimuli.

## 4. Materials and Methods

### 4.1. Identification of CML/CaM Genes in C. seticuspe

To identify *CML/CaM* gene family members in *C. seticuspe*, 57 CML/CaMs from *Arabidopsis thaliana* were collected as query sequences to search the *C. seticuspe* genome using BLASTP (v2.6.0+) with a cut-off e-value of <10^−15^ [27,29]. Candidate *CML/CaM* genes were collected. Genes were mapped to a SMART database (http://smart.embl-heidelberg.de/) (accessed on 7 September 2021) and genes without EF-hand domains were removed.

### 4.2. Phylogenetic Analysis of CsCaM/CML

To explore the evolution of CsCaM/CML, an alignment of multiple sequences of CsCaM/CMLs from *A. thaliana* and *C. seticuspe* was performed using MAFFT software (v7) [35]. The phylogenetic tree was constructed using an approximately maximum likelihood method via FastTree (v2.1.11) software [36]. The protein sequences of CaM/CML from *A. thaliana* and *C. seticuspe* were provided in Appendix A.

### 4.3. Gene Structure Construction and Protein Motif Prediction of CsCaM/CML

The coding sequences and the genomic sequences of *CsCaM/CML* were used to explore the gene structures of the *CsCaM/CML* gene family. The sequences were provided in Appendix A. The online website Gene Structure Display Server 2.0 (http://gsds.gao-lab.org/) (accessed on 15 September 2021) was used to depict the gene structure of gene family members based on the sequences submitted. The motif sites of CsCaM/CML protein were discovered using the online tool Multiple Em for motif elicitation (https://meme-suite.org/meme/tools/meme) (accessed on 20 September 2021) according to the CsCaM/CML protein sequences submitted [37].

### 4.4. Cis-Acting Regulatory Elements Analysis and Chromosomal Location

Upstream sequences (2000 bp) of *CsCaM/CML* gene family members were obtained from the *C. seticuspe* genome published recently [29]. The sequences were provided in Appendix A. The online database PlantCARE (http://bioinformatics.psb.ugent.be/webtools/plantcare/html/) (accessed on 23 September 2021) was used to predict the cis-acting regulatory elements of *CsCaM/CML* promoter sequences. Based on the position information of *CsCaM/CML* members in the genome, the chromosomal location of *CsCaM/CML* was physically mapped to each chromosome and depicted with MapChart tools (v2.3.2) [38].

### 4.5. Collinearity and Duplicate Events Analysis of CaM/CML

Protein sequences of *A. thaliana* and *C. seticuspe* were used to perform collinearity analysis; for genes with multiple transcripts, the longest transcripts were used. The proteins of *C. seticuspe* were mapped to *A. thaliana* and *C. seticuspe* using BLASTP with cutoff of e-value < 10^−5^ (hits were restricted to top 5). Collinearity blocks between *A. thaliana* and *C. seticuspe* and in *C. seticuspe* were analyzed using MCScanX software with default parameters [39]. Duplicate events of CMLs in *C. seticuspe* were analyzed using the ‘duplicate_gene_classifier’ function in the MCScanX software and the results were visualized using circos (http://circos.ca/) (accessed on 27 September 2021) [39]. To examine potential origins of duplications of *CMLs*, enrichment analysis was performed using the “origin_enrichment_analysis” function in the MCScanX package.

### 4.6. Expression Analysis of CsCaM/CML in Response to Abiotic Stress

To explore the response of *CsCaM*/*CML* under cold and salt stress, the RNA-seq data were downloaded from the ENA database. The buds were raised for 30 days under controlled conditions. For salt stress (PRJNA472473), samples were watered with 100 mM salt water; the same amount of purified water was used as control. Root samples were collected after 24 h of treatment. For low-temperature stress (PRJNA481579), the seedlings were treated with 4 °C for 24 h and then treated with −4 °C for 4 h. Plants in normal conditions were used as the control; leaves were collected for sequencing. Each experiment contained 6 samples—3 biological replicates in each condition (cold acclimation, salt treatment, and normal growth). The trimming tool Trimmomatic (v0.39) was used to trim the sequences in the sequencing files [40]. The reference genome of *C. seticuspe* (Gojo-0) was downloaded from PlantGarden (https://plantgarden.jp/) (accessed on 30 September 2021). Trimmed files were mapped to the reference genome using HISAT2 (v2.2.1) software with default parameters [41], and the gene expression of each sample was calculated using HTSeq (v0.11.5) software [42]. Files containing read counts of genes were merged. Different statuses of genes between two conditions were calculated with the edgeR package (v3.30.3) [43]. Genes with |logFC| of >1.5 and FDR of <0.05 were considered differentially expressed genes (DEGs); DEGs were depicted with volcano plots and a heatmap using the R software.

## 5. Conclusions

Based on the published genome of *C. seticuspe*, 86 *CsCaM/CML* genes were identified and classified into nine subgroups. The size of the *CML* family in *C. seticuspe* was significantly enlarged via whole-genome (or segmental) duplications. *CsCMLs* represented significant responses to cold and salt stress. Four significantly up-regulated *CMLs* (*CsCML2*/*3*/*9*/*80*) and 11 significantly down-regulated *CMLs* (*CsCML5*/*14*/*15*/*16*/*18*/*23*/*40*/*50*/*65*/*75*/*79*) were identified under cold treatment. Thirteen *CsCMLs* (*CsCML14/34/38/45/50/60/65/67/69/70/79/81*/*56*) were statistically significantly regulated in response to salt stress. Our findings provide a basis and foundation for exploring the roles of *CaM/CML* genes in the development of *C. seticuspe* and their functions in response to diverse environmental stimuli and stresses.

## Figures and Tables

**Figure 1 plants-11-01760-f001:**
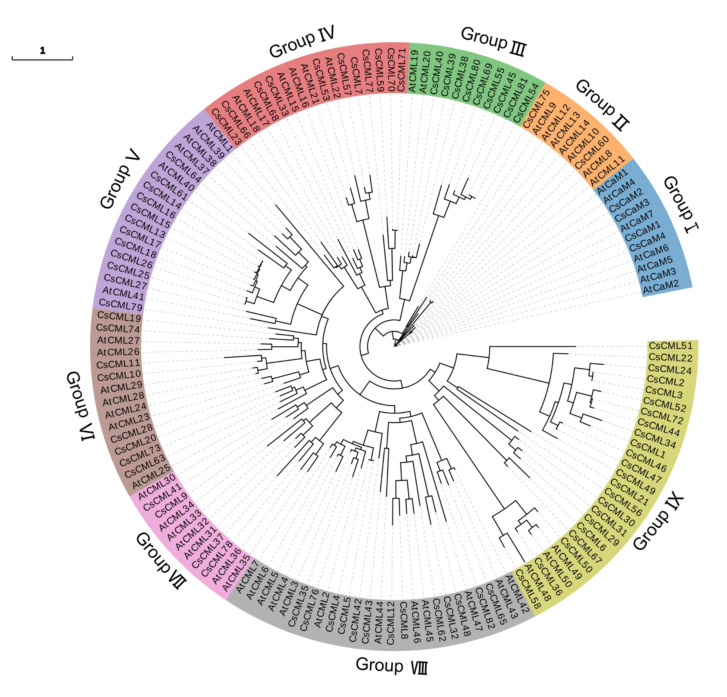
The phylogenetic tree of CsCaM/CML and AtCaM/CML. The phylogenetic analysis was constructed using FastTree with 57 CaM/CML in *Arabidopsis thaliana* and 86 CaM/CML in *Chrysanthemum seticuspe*; different groups were color-coded.

**Figure 2 plants-11-01760-f002:**
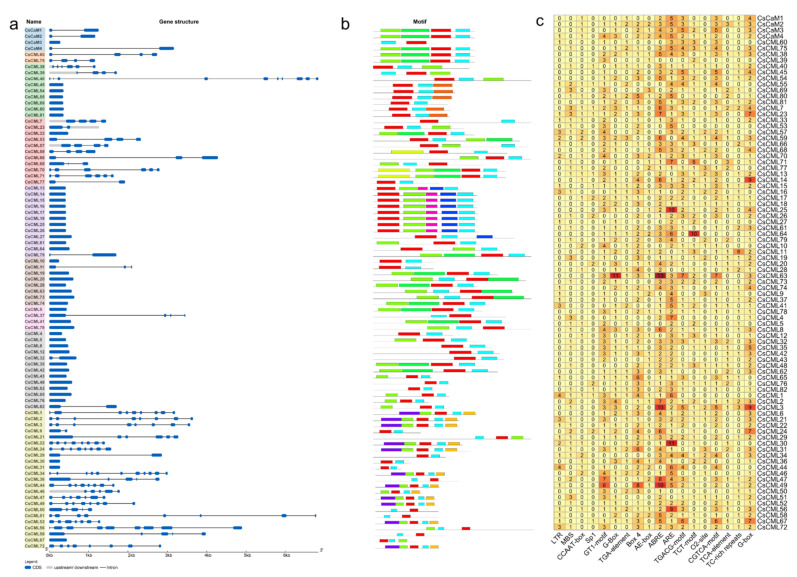
Sequences analysis of *CsCaM/CML* genes. Gene structures (**a**), motifs (**b**), and cis-acting regulatory elements (**c**) of *CsCaM/CML* gene family members were analyzed and depicted.

**Figure 3 plants-11-01760-f003:**
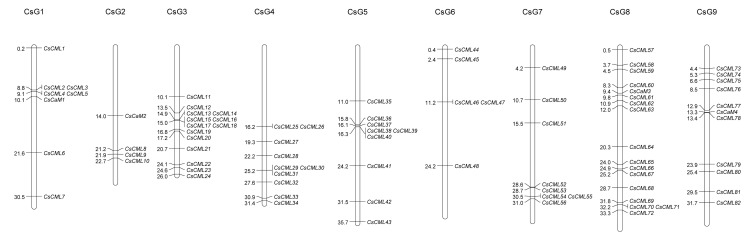
Chromosomal locations of *CsCaM/CML* members. Distributions of *CsCaM/CML* genes on *Chrysanthemum seticuspe* chromosomes were depicted.

**Figure 4 plants-11-01760-f004:**
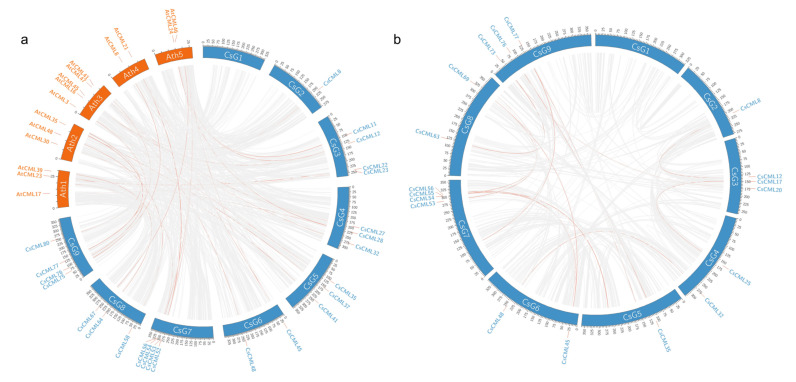
Collinearity analysis of *CsCaM/CML* genes. Circos plots displayed collinearity blocks between *Arabidopsis thaliana* and *Chrysanthemum seticuspe* (**a**) as well as within *Chrysanthemum seticuspe* (**b**). Links containing *CsCaM/CML* were colored with red; *CaM/CML* genes involved in collinearity blocks were labeled.

**Figure 5 plants-11-01760-f005:**
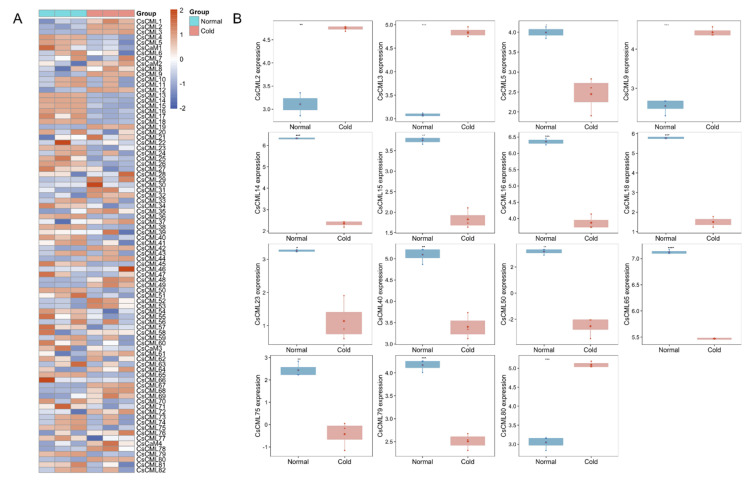
Expression of *CsCaM/CML* under low-temperature stress. (**A**) The heatmap showed the expression of *CsCaM/CML* gene family members in response to cold stress; gene expressions were scaled in the row. (**B**) Significantly differentially (*) expressed genes were depicted with boxplots using transcripts per million (TPM). Significance level symbols with one, two, three, and four asterisks represent *p* values less than 0.05, 0.01, 0.001 and 0.0001, respectively.

**Figure 6 plants-11-01760-f006:**
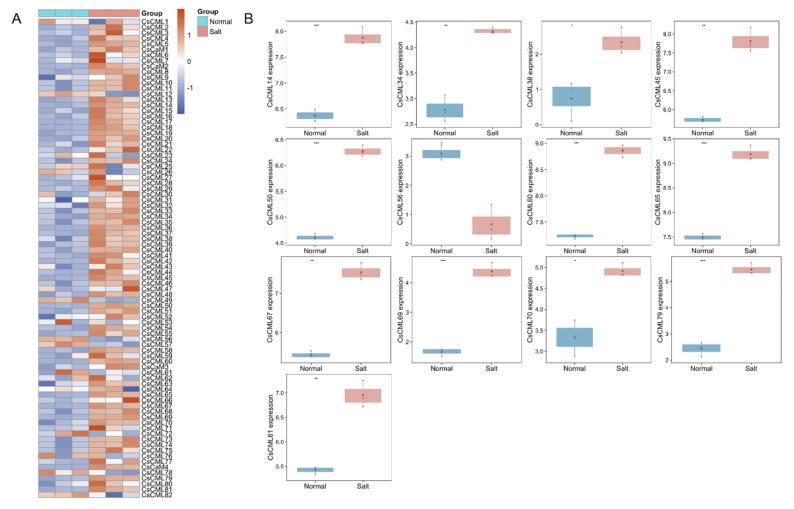
Expression of CsCaM/CML genes under salt stress. (**A**) Gene expression levels of CsCaM/CML gene family member response to salt stress were depicted with the heatmap; gene expressions were scaled in the row. (**B**) Significantly differentially expressed genes under salt treatment were depicted with boxplots using TPM. Significance level symbols with one, two, three, and four asterisks represent *p* values less than 0.05, 0.01, 0.001 and 0.0001, respectively.

## Data Availability

Not applicable.

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
