# Peer review of "Genome-Wide Identification and Expression Analysis of CsCaM/CML Gene Family in Response to Low-Temperature and Salt Stresses in Chrysanthemum seticuspe"

_plants, 2022, doi:10.3390/plants11131760_

Round 1
Reviewer 1 Report
The present work encompasses identification of candidates genes pertaining to low temperature and salt stress in Chrysanthemum seticuspe. The manuscript is well written and results were precisely explained. However, I would suggest some changes required for the improvement of the manuscript.
1. Orthologous gene clusters identification needs to be concluded.
2. Validation of candidate genes by qRT-PCR is required and results are needed to be incorporated in the manuscript.
3. Conclusions should be precisely written along with the names of predominant stress associated genes (validated) along with the proposed mechanism.
Reviewer 2 Report
In this paper submitted to Plants, Manman Fu and colleagues report a Genome wide identification of CaM/CML genes in Chrysanthemum seticuspe (Cs), as well as a phylogenetic analysis of the CaM/CML family, a description of gene structures and protein motifs, a prediction of cis-acting regulatory elements and an analysis of gene expression in response to low-temperature and salt stresses. This study based on publicly available data (including genome assembly for Cs and RNA-seq data in response to abiotic stress) was well designed and the paper is well written. I have identified only minor shortcomings that the authors should consider to improve their manuscript:
Introduction
1) This section is relatively short and the authors could add some details. For example, they could elaborate on what “EF-hand patterns” are (L35), or explain what they mean by “Ca2+ was stimulated” L29.
2) L38, the authors could use the abbreviation for absisic acid (given in the abstract, L17)
3) The first sentence of the paragraph L45-46 is misplaced. It can be moved and merged with the sentence that starts L53.
Materials and Methods
4) They authors should provide the gene IDs for the 57 CaM/CML from Arabidopsis thaliana (At) (as supplementary material for example). Also, contrary to what is written, the supplementary file 1 doesn’t contain the protein sequences of the 57 CaM/CML from At, but only the sequences from the 86 proteins identified in Cs in this study.
5) L80: a space is missing after the dot following the reference #31.
6) L98: a space is missing between “tools” and the following parenthesis.
7) L101-102: in the analysis of collinearity between At and Cs, a reciprocal best hit BLAST approach would have been preferable to a simple mapping using BLASTP
8) I think that the sentence L119-120 on Trimmomatic should be improved.
9) In general, in the M&M, the authors should provide versions of each bioinformatics tool.
Results
10) In general, figures are of poor quality (resolution) and are difficult to visualize.
- For example, Figure 2 will be too small with this layout. Perhaps part C could be placed below parts A and B, so as to increase the font size of the labels in part A and the numbers in part C.
- Figure 4 is really too small.
11) I think Table 1 It is not very useful in the text. It would be better to give it as supplementary material, so that readers can download it and work directly in a spreadsheet.
